# A Sequential Color Correction Approach for Texture Mapping of 3D Meshes

**DOI:** 10.3390/s23020607

**Published:** 2023-01-05

**Authors:** Lucas Dal’Col, Daniel Coelho, Tiago Madeira, Paulo Dias, Miguel Oliveira

**Affiliations:** 1Intelligent System Associate Laboratory (LASI), Institute of Electronics and Informatics Engineering of Aveiro (IEETA), University of Aveiro, 3810-193 Aveiro, Portugal; 2Department of Mechanical Engineering (DEM), University of Aveiro, 3810-193 Aveiro, Portugal; 3Department of Electronics, Telecommunications and Informatics (DETI), University of Aveiro, 3810-193 Aveiro, Portugal

**Keywords:** pairwise-based color correction, texture mapping, joint image histogram, color mapping function, weighted graphs

## Abstract

Texture mapping can be defined as the colorization of a 3D mesh using one or multiple images. In the case of multiple images, this process often results in textured meshes with unappealing visual artifacts, known as texture seams, caused by the lack of color similarity between the images. The main goal of this work is to create textured meshes free of texture seams by color correcting all the images used. We propose a novel color-correction approach, called sequential pairwise color correction, capable of color correcting multiple images from the same scene, using a pairwise-based method. This approach consists of sequentially color correcting each image of the set with respect to a reference image, following color-correction paths computed from a weighted graph. The color-correction algorithm is integrated with a texture-mapping pipeline that receives uncorrected images, a 3D mesh, and point clouds as inputs, producing color-corrected images and a textured mesh as outputs. Results show that the proposed approach outperforms several state-of-the-art color-correction algorithms, both in qualitative and quantitative evaluations. The approach eliminates most texture seams, significantly increasing the visual quality of the textured meshes.

## 1. Introduction

Three-dimensional reconstruction can be summarized as the creation of digital 3D models from the acquisition of real-world objects. It is an extensively researched topic in areas such as robotics, autonomous driving, cultural heritage, agriculture and medical imaging. In robotics and autonomous driving, 3D reconstruction can be used to obtain a 3D perception of the car’s surroundings and is crucial to accomplish tasks such as localization and navigation in a fully autonomous approach [1,2,3]. In cultural heritage, 3D reconstruction aids in the restoration of historical constructions that have deteriorated over time [4]. In agriculture, 3D reconstruction facilitates the improvement of vehicle navigation, crop, and animal husbandry [5]. In medical imaging, reconstruction produces enhanced visualizations which are used to assist diagnostics [6].

The technologies used to reconstruct a 3D model are typically based on RGB cameras [7], RGB-D cameras [8,9], and light detection and ranging (LiDAR) [10,11]. Point clouds from LiDAR sensors contain quite precise depth information, but suffer from problems such as occlusions, sparsity, and noise, and images from RGB cameras provide color and high-resolution data, but no depth information [12]. Nonetheless, the fusion of LiDAR point clouds and RGB images can achieve better 3D reconstruction results than a single data modality [13] and, for this reason, is often used to create textured 3D models.

Texture mapping is the colorization of a 3D mesh using one or more images [14,15,16,17]. In the latest applications, usually several overlapping images are available to texture the 3D mesh, and a technique to manage the redundant photometric information is required. Those techniques are often referred to as multiview texture mapping [18,19,20,21]. The winner-take-all approach is usually the technique used to handle the redundant photometric information. For each face of the 3D mesh, this approach selects a single image from the set of available images to colorize it. The selection of the image raises the problem of how to select the most suitable one from the set of available images and how to consistently select the images for all faces of the 3D mesh. To address those problems, [22] employ a Markov random field to select the images. The authors of [23] use graph optimization mechanisms. Others use optimization procedures to minimize the discontinuities between adjacent faces [24]. Another approach that is commonly used is the average-based approach, which fuses the texture from all available images to color each face. The authors of [25,26] propose the fusion of the textures based on different forms of weighted average of the contributions of textures in the image space. However, these approaches are highly affected by the geometric registration error between the images, causing blurring and ghost artifacts, which are not visually appealing [27]. The winner-take-all approach has the advantage of solving the problem of blurring and ghost artifacts. However, particularly in the transitions between selected images, texture seams are often perceptible. These texture seams are caused by the color dissimilarities between the images. To tackle this additional problem, robust color correction techniques are often used to color correct all images, in an attempt to create the most seamless textured 3D meshes possible. Color correction can be defined as the general problem of compensating the photometrical disparities between two coarsely geometrically registered images. In other words, color correction consists of transferring the color palette of a source image **S** to a target image **T** [28].

There are several color-correction approaches in the literature; however, the majority of them focus on correcting a single pair of images, whereas our objective is to increase the level of similarity between multiple images from a 3D model. Nonetheless, in the past few years, the color correction across image sets from the same scene has become a widely researched subject [29,30,31]. Apart from the different techniques to ensure the color consistency between multi-view images, color correction for both image pairs and for image sets can be defined as the same problem of adjusting the color of two or more images in order to obtain photometric consistency [31]. The methods of color correction across image sets are usually formulated as a global optimization problem to minimize the color discrepancy between all images [30]. On the other hand, the methods of color correction for image pairs are usually designed to transfer the color palette of a source image **S** to a target image **T** and are only used to color correct a single pair of images [28]. Nevertheless, we believe that pairwise-based methods can be extended to address the problem of color correcting multiple images of a scene, which is usually the case for texture mapping applications. Pairwise-based methods are naturally less complex and easier to implement than optimization methods. However, the extension of these methods to color correct several overlapping images is not straightforward.

In this light, we propose a novel pairwise-based approach that uses a clever sequence of steps, allowing for a pairwise method that was originally designed to color correct a single pair of images, to color correct multiple images from a scene. The manner in which we carry out the sequence of pairwise steps is the key to achieving an accurate color correction of multiple images, as will be detailed in Section 3.

The remainder of the paper is structured as follows: Section 2 presents the related work, concerning the main color correction approaches available in the literature and how we can contribute to the state-of-the-art; Section 3 describes the details of the proposed color correction approach; Section 4 discusses the results achieved by the proposed approach, and provides a comparison with other state-of-the-art approaches; and finally, Section 5 presents the conclusions and future work.

## 2. Related Work

There are two main color correction approaches: based on image pairs, or using a larger set of images. In this section, we start by reviewing image pairs color-correction techniques. Then we present techniques of color correction for image sets. At the end, a critical analysis of the state-of-the-art and a contextualization of our work is presented. Regarding color correction based on image pairs, existing techniques are divided in two classes: model-based parametric and model-less non-parametric. Model-based uses the statistical distribution of the images to guide the correction process. The authors of [32] were the pioneers with a linear transformation to model the global color distribution from a source image **S** to a target image **T**. They used Gaussian distributions (mean and standard) to correct the source image color space according to the target image. This approach is commonly used as a baseline for comparison [33,34,35]. Another approach [34] used the mean and covariance matrix to model the global color distribution of the images in the RGB color space. To estimate the color distribution more accurately, the authors of [36] proposed the usage of Gaussian mixture models (GMMs) and expectation maximization (EM) rather than simple Gaussian. They also proposed a local approach by modeling the color distribution of matched regions of the images. This local correction is also used in [37], where the mean shift algorithm was used to create spatially connected regions modeled as a collection of truncated Gaussians using a maximum likelihood estimation procedure. Local color correction approaches can better handle several reasons for differences between images: different color clusters in the same image, differing optics, sensor characteristics, among others [38]. A problem that may appear when correcting the three image channels independently are cross-channel artifacts. To avoid these artifacts, 3D GMMs can be used to model the color distribution in all three channels [35] or multichannel blending [39,40] since they model the three color channels simultaneously.

Model-less non-parametric approaches are usually based on joint image histogram (JIH) from the overlapped regions of a pair of images. A JIH is a 2D histogram that shows the relationship of color intensities at the exact position between a target image **T**, and a source image **S**. Then, a JIH is used to estimate a color mapping function (CMF): a function that maps the colors of a source image **S** to a target image **T** [28]. However, CMFs are highly affected by outliers that may appear due to camera exposure variation [41], vignetting effect [42], different illumination, occlusions, reflection properties of certain objects, and capturing angles, among others [28]. The radiometric response function of the camera may also affect the effectiveness of the CMF in color correcting a pair of images. To reduce this problem, the monotonicity of the CMF has to be ensured using, for example, dynamic programming [43]. In [44,45], the CMF estimation is performed with a 2D tensor voting approach followed by a heuristic local adjustment method to force monotonicity. The same authors estimated the CMF using a Bayesian framework in [46]. Other possibilities for CMF estimation are energy minimization methods [47], high-dimensional Bezier patches [48] or the use of a Vandermonde matrix to model the CMF as a polynomial [49]. A root-polynomial regression, invariant to scene irradiance and to camera exposure, was proposed in [41]. A different approach corrects images in the RGB color space with a moving least squares framework with spatial constraints [50]. Color-correction approaches for image pairs are not straightforward to extend to image set that uses three or more images for the correction.

Most color-correction approaches for image sets are based on global optimization algorithms that minimize the color difference of the overlapping regions to obtain a set of transformation parameters for each image. These approaches are common to several applications, such as image stitching, and 3D reconstruction, among others. The authors of [39] addressed the problem with a global gain compensation to minimize the color difference in the overlapped regions. This approach was proposed for image stitching and has is implemented in the *OpenCV stitching model* https://docs.opencv.org/4.x/d1/d46/group__stitching.html (accessed on 19 November 2022) and in the panorama software *Autostitch* http://matthewalunbrown.com/autostitch/autostitch.html (accessed on 19 November 2022). In [51], a linear model was used to estimate independently three global transformations for each color channel, minimizing the color difference through histogram matching. The overlapped regions were computed using image-matching techniques, such as scale-invariant feature transform (SIFT) key points [52]. This method was implemented in *OpenMVG* https://github.com/openMVG/openMVG (accessed on 19 November 2022) [53]. Linear models might not be flexible enough to minimize high color differences between regions of the images. In this context, the authors of [54] proposed a gamma model rather than a linear model to avoid the oversaturation of luminance by applying it in the luminance channel in the YCrCb color space. In the other two chromatic channels, the authors maintained the linear model through gain compensation.

The approaches described in the previous paragraph apply the least-square loss function as the optimization solver, but this method is very sensitive to outliers and missing data. An alternative proposed in [29] estimated the gamma model parameters through low-rank matrix factorization [55]. Although the linear and gamma models achieved good results, they present some limitations with challenging datasets with high color differences, resulting in the use of other models. A quadratic spline curve, more flexible to correct significant color differences, was proposed in [56]. The same quadratic spline curves were used in [30] with additional constraints concerning image properties, such as gradient, contrast, and dynamic range. In [31], the authors estimated a global quadratic spline curve by minimizing the color variance of all points generated by the structure from motion (SfM) technique [57]. The robustness of this method was improved with the adoption of strong geometric constraints across multiview images. Moreover, to improve efficiency through large-scale image sets, the authors proposed a parallelizable hierarchical image color correction strategy based on the m-ray tree structure.

The previous paragraphs have detailed several approaches to deal with the problem of color correcting a pair of images or multi-view image sets. Many approaches have been proposed that aim to color correct image sets through global optimization processes. However, only a few of them are directed to the problems of texture mapping applications. For the case of indoor scenarios, these problems include the following: (i) overlapped regions with a very different size because of the varying distance from the cameras to the captured objects; (ii) considerable amounts of occlusions; and (iii) lack of overlapped regions. In recent papers for color correction in texture mapping applications, [31,58] tackle the problem of texture mapping for outdoor scenarios, which do not have many of the issues discussed above, such as overlapped regions with very different sizes and occlusions. On the other hand, color-correction approaches for image pairs are simpler methods, but are designed to color correct only a single pair of images, making it difficult to generalize these approaches to color correct multiple images of a scene. However, we believe that color-correction algorithms for image pairs can be adapted to color correct multiple images for two reasons: (i) these algorithms are simpler and more efficient methods because only two images at each step are handled by the algorithm, and in contrast, the optimization-based algorithms try to deal with the color inconsistencies of all images at the same time; and (ii) these algorithms could be executed in an intelligent sequence of steps across the set of images in order to increase the color consistency between all images. By using a pairwise method, we take advantage of all the years of research and all the approaches developed to color correct an image pair.

Our previous work [59] has focused on the usage of 3D information from the scene, namely point clouds and 3D mesh, to improve the effectiveness of the color correction procedure, by filtering incorrect correspondences between images from an indoor scenario. These incorrect correspondences are usually due to occlusions and different sizes of the overlapped regions and may lead to poor performance of the color correction algorithm.

In this paper, we propose a novel pairwise-based color correction approach adapted to solve the problem of color correcting multiple images from the same scene. Our approach aims to use color-correction algorithms for image pairs since we believe that, in some aspects, optimization-based methods may become overly complicated to color correct several images from large and complex scenes. The proposed approach uses a robust pairwise-based method in a clever sequence of steps, taking into consideration the amount of photometric information shared between the pairs of images. This sequential approach color corrects each image with respect to a selected reference image increasing the color consistency across the entire set of images.

## 3. Proposed Approach

The architecture of the proposed approach is depicted in Figure 1. As input, the system receives RGB images, point clouds from different scans, and a 3D mesh reconstructed from those point clouds. All these are geometrically registered with respect to each other. Our approach consists of 8 steps. In the first step, the faces of the 3D mesh are projected onto the images, to **compute the pairwise mappings**. Pairwise mappings are corresponding pixel coordinates from the same projected face vertices, in a pair of images. Then, several techniques are applied to **filter incorrect pairwise mappings** that would undermine both color-correction and texture-mapping processes. Once the incorrect pairwise mappings have been removed, a **color correction weighted graph is created**, containing all the pairs of images available to use in the color correction procedure. A **color correction path is computed** for each image using a path selection algorithm between each of those images and a reference image that must be selected by the user. Subsequently, we **compute a joint image histogram (JIH)** for each pair of images within the color correction paths and **estimate a color mapping function (CMF)** that best fits each JIH data. To finish the color correction procedure, we perform what we call **sequential pairwise color correction**, by using the color correction paths and the CMFs, to effectively color correct all images with respect to the reference image. At the end of this stage, the corrected RGB images are produced and then used in the last step to colorize each face of the 3D mesh through a **texture mapping image selection** algorithm, resulting in the textured 3D mesh. The information from the pairwise mappings filter is also used to increase the robustness of the image selection technique.

This paper focuses on three components of the pipeline: color correction weighted graph creation (Section 3.3), color correction paths computation (Section 3.4), and sequential pairwise color correction (Section 3.7). Other components, such as pairwise mappings computation and pairwise mappings filtering, have been addressed in detail in our previous work [59]. In this context, we consider it important for the reader to have an overview of the complete framework, which is why we offer a less detailed explanation of these procedures in Section 3.1 and Section 3.2. Section 3.3, Section 3.4 and Section 3.7 describe the core contributions of this paper in detail.

To produce the results and show the entire pipeline, we used a dataset from a laboratory at IEETA — Institute of Electronics and Informatics Engineering of Aveiro in University of Aveiro. This dataset represents an entire room and contains 24 images from different viewpoints, 9 point clouds from different scans, and a 3D mesh reconstructed from those point clouds with 41,044 faces and 21,530 vertices representing the entire room.

### 3.1. Computation of Pairwise Mappings

The geometric registration between a pair of images is usually determined based on overlapping areas, which contain photometrical information between regions of pixels that may or may not correspond to the same object. The usage of 3D information, namely 3D meshes and point clouds, to compute the correspondences between images, can be more accurate and reliable than image-based techniques, especially in indoor scenarios, which are more cluttered and contain more occlusions.

As discussed above, our previous work [59] initially proposed to use 3D meshes to compute correspondences between image pairs. For this reason, the proposed approach starts with the computation of the pairwise mappings that takes as input the RGB images and the 3D mesh. Firstly, the faces of the 3D mesh are projected onto the images. We make use of the pinhole camera model [60] to compute the projections of all vertices of the faces, resulting in pixel coordinates of the image for each projected vertex. The projection of a face onto an image is valid when two conditions are met: (i) all vertices of the face must be projected inside the width and the height of the image; and (ii) the *z* component of the 3D coordinates of the face vertices, transformed to the camera’s coordinate reference system, must be greater than 0 to avoid the projection of vertices that are behind the camera.

After the computation of the valid projections onto all images, we compute the pairwise mappings. Pairwise mappings are pixel coordinates from a pair of images that correspond to the same projected vertices. For every pair of images, we evaluate, for each face, if the projection of that face is valid in both images. For each valid face in those conditions, we create a pairwise mapping for each vertex of that face. One pairwise mapping is represented by the pixel coordinates of the vertex projection onto the first image and onto the second image. In Figure 2, the pairwise mappings are illustrated using the same color to showcase corresponding mappings between a pair of images.

### 3.2. Filtering of Pairwise Mappings

It is noticeable that the pairwise mappings at this stage of the pipeline contain noisy data (see Figure 2) caused by the problems mentioned in Section 2: occlusions, reflection properties of certain objects, and capturing angles, among others. For example, in Figure 2a, there are mappings represented in blue and green colors that represent the ground at the center of the room. In Figure 2b, the same mappings represent the center table. This example demonstrates inaccurate pairwise mappings between two different 3D objects. These inaccurate associations introduce incorrect photometrical correspondences between a pair of images, which would disrupt the color-correction procedure. Occluded faces and registration errors are normally the cause of inaccurate pairwise mappings, especially in indoor scenarios. A face is considered occluded from a camera point of view in the following cases: (a) the occluding face intersects the line of sight between the camera and the occluded face; and (b) the occluding face is closer to the camera. Regarding registration errors, we used a high-precision laser scanner that guarantees a very low average registration error. However, in situations where the angle between the camera viewpoint (focal axis of the camera) and the normal vector of the face is excessively oblique, or in other words, close to $90^{\circ }$, the impact of the registration error on the accuracy of the pairwise mappings is amplified, increasing the possibility of inaccurate pairwise mappings. In this paper, we leverage previous work [59], which uses a filtering procedure composed of a combination of 3 filters: z-buffering filtering, depth consistency filtering and camera viewpoint filtering. Each pairwise mapping is only considered valid if it passes the three filtering steps.

The z-buffering filtering is used to discard the occluded faces from an image point of view. This filter evaluates all faces to assess if they are occluded or not by other faces, considering that point of view. A face is not occluded by another face when considering an image point of view, when one of two conditions are satisfied: (i) the projection of the vertices of the evaluated face does not intersect the projection of the vertices of the other face, i.e., the faces do not overlap each other; (ii) or, in case that there is an intersection between the faces, the maximum Euclidean distance of the three vertices of the evaluated face must be less than the minimum Euclidean distance of the three vertices of the other face, i.e., the evaluated face is in front of the other face and, therefore, is not occluded. This filter is computed for each viewpoint, i.e., for each image. Figure 3 depicts the entire filtering procedure applied in the same images shown in Figure 2 from the original pairwise mappings to the filtered pairwise mappings, illustrating the impact of each filter to eliminate the noisy data. Figure 3c,d show the pairwise mappings with only the z-buffering filter applied. It is possible to observe that in Figure 3c, the mappings on the ground at the center of the room were discarded because in Figure 3d, they are occluded by the center table.

The depth consistency filtering aims to discard the remaining occlusions due to discrepancies between the mesh and the point cloud. These discrepancies can have multiple sources, such as mesh irregularities, non-defined mesh, and registration issues, among others. This filter estimates the distance from the camera pose to the center of the face, or in other words, the depth of the face, with two different methods. The first method computes the depth of the face based on the 3D mesh. This is done by computing the $L_{2}$ norm of the vertices coordinates with respect to the camera, and then selecting the minimum value. The second method computes the depth of the face based on the point cloud. This is done by using the partial point clouds from the setups where the images were taken to project the depth information and then create depth images. However, pixels without information occur due to the sparse points of the point cloud. An image inpainting process based on the Navier–Stokes equations for fluid dynamics is used to fill the missing parts of each depth image using information from the surrounding area [61]. The depth can be directly extracted from the depth images using the coordinates obtained from the vertex projection. Finally, the algorithm compares the difference between those depth values to find depth inconsistencies and discard the inaccurate pairwise mappings. This is computed for all faces from each camera viewpoint. Figure 3e,f show the pairwise mappings with both the z-buffering filter and the depth consistency filter applied. By comparing them with Figure 3c,d, this filter discarded the mappings on the legs of the tables with respect to the wall under the table. The reason for this is that the legs of the tables are not defined on the mesh due to their reduced thickness, and yet the filter was able to discard those incorrect pairwise mappings. Additionally, the pairwise mappings on the table against the wall were discarded since the tables are not well-defined on the mesh due to difficulties in correctly acquiring both the top and bottom surfaces of the table.

The last filter is the camera viewpoint filtering that aims to remove pairwise mappings in which the angle between the camera viewpoint (focal axis of the camera) and the normal vector of the face is excessively oblique. When this occurs, the impact of the registration error on the accuracy of those pairwise mappings is amplified. For this reason, these pairwise mappings are more likely to be incorrect, and therefore should be discarded. Figure 3g,h show the pairwise mappings with all the three filtering methods applied. Regarding the camera viewpoint filtering, we can observe that the remaining pairwise mappings on the center table were discarded due to its excessive oblique angle relative to the camera viewpoint in Figure 3g. Additionally, the discarded mappings on the ground have an excessive oblique angle relative to the camera viewpoint in Figure 3h. Figure 3h clearly shows the extensive amount of noise in the dataset used. This reinforces the importance of using 3D information not only to compute the correspondences between images, but to discard the incorrect ones.

### 3.3. Creation of Color-Correction Weighted Graph

For texture mapping applications, especially in indoor scenarios, there are many pairs of images that do not overlap or have minimal overlap, as described in Section 2. Those pairs share none or insufficient photometrical information, to support an accurate color correction. Whenever possible, these pairs should be avoided, and not used in the color-correction procedure. The number of pairwise mappings computed previously can be used as an indicator of this shared photometrical information between each pair of images, as detailed in Section 3.1 and Section 3.2. Therefore, a threshold using the number of pairwise mappings $t_{npm}$ can be used to discard the unfavorable pairs from the color correction procedure. By discarding these, the algorithm only carries out color correction between pairs that share enough photometrical information, leading to a more accurate color correction procedure. Furthermore, the higher the number of pairwise mappings, the higher the amount of shared photometrical information between a pair of images.

The color-correction procedure in this paper is pairwise-based. Hence, the information about all the pairs of images available to carry out the color-correction procedure, according to the criterion described above, should be represented in a data structure that supports the creation of color-correction paths between images. In this light, we propose the creation of a color correction graph *G*, where the nodes represent the individual images, and the edges represent all the pairs of images, which were not filtered, according to the threshold $t_{npm}$. *G* is represented as follows:(1)G={V,E,w},
where $V=\{I_{1},I_{2},...,I_{n}\}$ are the graph nodes representing the individual images, $E\subseteq \{\left\{i,k\right\}\;|\;i,k\in V\;and\;i\neq k\;\;and\;n\left(M_{\langle i,k\rangle }\right)\geq t_{npm}\}$ are the graph undirected edges representing the pairs of images available to the color-correction procedure and w:E→Q are the edge weight functions. $n(M_{\langle i,k\rangle })$ is the cardinality of the set that contains the pairwise mappings associated with the *i-th* and *k-th* images. It would be helpful that each pair of images contains a color-correction quality indicator that represents how suitable that pair is to contribute to a successful color-correction procedure. For example, a path-selection algorithm can use this color correction quality indicator, to decide the sequence of images to transverse, in order to produce an accurate color correction. The number of pairwise mappings seems to be an appropriate indicator of an accurate color correction for each pair of images. As such, we use it as the weights of the graph. The weights are normalized according to the total number of pairwise mappings as follows:(2)$$w_{ik}=1-\frac{n(M_{\langle i,k\rangle })}{\sum n(M_{\langle i,k\rangle })}\;,$$
where $w_{ik}$ is the edge weight associated with the *i-th* and *k-th* images, and $\sum n(M_{\langle i,k\rangle })$ is the sum of the cardinality of all sets of pairwise mappings.

Figure 4 shows an example of the color correction weighted graph for a subset containing six images as the nodes of the graph. The arrows represent the edges of the graph and the thickness of the edge lines represents their cost, which means that a better path will transverse thinner arrows. For example, the edge that connects the pair of images (**F**, **B**) has a low cost (thinnest arrow), and it is possible to observe that this pair contains a high amount of pairwise mappings, because most of the objects are visible in both images, such as the posters, the wall, and the table. On the other hand, between the pair of images (**B**, **A**) there are only small common regions, resulting in a high cost (thickest arrow).

### 3.4. Computation of Color-Correction Paths

We propose to select one image from the set of images as the reference image. This reference image is used as the color palette model to guide the color-correction procedure. The selection is arbitrary and is left to the user. Section 3.7 will provide details on how this selection is carried out and the advantages of this approach. With the color-correction weighted graph created in the previous section, the information about all the pairs of images available to carry out the color-correction procedure is structured in a graph. In the graph, there are images that have a direct connection (edge in the graph) with the selected reference image. In these cases, the algorithm can perform a direct color correction. On the other hand, in most cases, there are images in the graph that do not have a direct connection to the reference image. In these cases, it is not possible to perform a direct color correction. To overcome this, we propose to compute color-correction paths from each image in the set to the reference image. These paths are automatically computed from the color-correction weighted graph, described in Section 3.3. We chose the Dijkstra’s shortest path algorithm [62] to find the path with the minimum cost. For this reason, the computed color-correction path for each image will have the highest sum of pairwise mappings, thus increasing the accuracy of the pairwise color correction.

We use the color-correction paths to color correct all images with reference to this reference image, resulting in images more similar in color to the reference image and also with higher color consistency between each other. Section 3.7 will provide details on how the color correction of each image of the dataset is performed through the computed color correction paths. It is noteworthy that every image from the dataset should have at least one edge in the graph, otherwise the path selection algorithm will not be able to compute. Figure 5 shows the path computed by the algorithm from image **E** (in green) to the reference image **F** (in red). Image **E** has no connection (edge) with the reference image **F**. However, the path selection algorithm selects an adequate path that goes through image pairs with high overlap.

### 3.5. Computation of Joint Image Histograms

A joint image histogram (JIH) is a 2D histogram created from the color observations of the pairwise mappings between a target image **T**, and a source image **S**. An entry in this histogram is a particular combination of color intensities for a given pairwise mapping. In this section, a JIH is computed for each channel of each pair of images in the computed color correction paths, using the pairwise mappings. A JIH is represented by JIH(x,y), where x and y represent all possible values of colors in **T** and **S**, being defined in the discrete interval $\left[0,2^{n}-1\right]$, where *n* is the bit depth of the images.

Figure 6 shows an example of a JIH of the red channel from a given pair of images. The red dots represent the observations according to the pairwise mappings, and the observation count is represented by the color intensity of each point, which means that the higher the color intensity, the higher the number of observations in that cell. To visualize the amount of noise coming from the pairwise mappings, gray dots are drawn to represent discarded pairwise mappings, but obviously are not used to estimate the CMFs. The JIHs are used to estimate the CMFs, which will then be used to transform the colors of each image from the dataset, thus producing the color-corrected images.

### 3.6. Estimation of Color Mapping Functions

A color mapping function (CMF) is a function that maps the colors of a source image **S** to a target image **T**. Using this function, the colors of the target image **T** are transformed, resulting in a color corrected image $\hat{T}$ which is more similar in color to the source image **S**. Each JIH created before is used as the input data to estimate a CMF. Since there are three JIHs for every pair of images combined from the color-correction paths, one for each channel of the RGB color space, there are also three CMFs.

The set of three CMFs, one for each color channel, is used to color correct all pixels of the input image. The RGB channels of the input image are color corrected independently by each one of the CMFs. In each step of the pairwise color correction, the CMF receives as input the color value x of each pixel of the target image **T** and then returns the color-corrected value $\hat{x}$. The original color value x of the pixel is transformed to the color-corrected value $\hat{x}$. After performing the color correction for all pixels of the target image and for each channel independently, the channels are merged to generate the corrected image $\hat{T}$. In this context, the CMF can be formulated as
(3)$$\begin{array}{c} \hat{x}=\hat{f}\left(x\right),\;\forall \;x\in \left[0,2^{n}-1\right] \end{array}\;,$$
where $\hat{f}$ is the estimated CMF for each JIH given as input data, and $\hat{x}$ is the resultant color of the color-corrected image $\hat{T}$ for a given color x of the target image **T**.

Since the CMF is a function, it cannot map a single element of its domain to multiple elements of its codomain. For the entire range of values, there is one and only one resulting corrected value $\hat{x}$ for each color of the target image x. However, for each of the values in x for a typical JIH, there are several observations of y. For this reason, it must be assumed that there is a considerable amount of noisy observations in a JIH.

This work proposes to estimate the CMF using a regression analysis to fit the color observations from the JIH. We created a composed model of the support vector regressor (SVR) [63] called composed support vector regressor (CSVR), which combines the linear kernel and the radius basis function kernel. This composed model was created because the radius basis function kernel is not able to extrapolate in the columns of the JIH where there are only a few or no observations, thus becoming very unpredictable in those columns and, most of the time, ruining the color-correction procedure. We estimate one SVR with the linear kernel and another SVR with the radius basis function kernel. Then, we discover the first intersection $x_{0}$ and the last intersection $x_{n}$ between the two functions. Subsequently, we apply the linear function for the intervals $x<x_{0}$ and $x>x_{n}$, and the radius basis function for the interval $x_{0}\leq x\leq x_{n}$. Figure 7 presents an example of the CMF estimation using the CSVR regression model for the given JIHs of a pair of images. The red, green, and blue curves represent the CMFs estimated for the red, green, and blue channels, respectively. For each channel, the CMF appears to fit the data of the JIH quite well, following the greater peaks of observations. The amount of noisy mappings (gray dots) eliminated by the filtering procedure is significant (see Figure 6 and Figure 7). Without the filtering procedure, the JIHs would be more dispersed, which would cause a negative impact on the effectiveness of the regression analysis.

We compute only the JIHs and the CMFs that will be used in the color-correction procedure, according to the computed color correction paths. As the number of images in a dataset increases, the combinations of pairs of images increase exponentially. For this reason, the computation burden is limited by not computing the JIHs and the CMFs for every pair of images in the dataset, as many of them are not used in the color-correction procedure.

### 3.7. Sequential Pairwise Color Correction

The objective of the color correction in an image pair, also called pairwise color correction, is to transform the colors of the target image, effectively making them more similar to the colors of the source image. In Section 3.4, we computed the color-correction paths between each image in the set and the reference image. We propose a sequential pairwise color correction. We use the CMFs estimated between the pairs of images formed sequentially along each color-correction path. Each path contains a list of images that must be crossed in order. The first image in the list is the image to be color corrected (target image), and the last image in the list is the reference image. We travel across the list, carrying out a color-correction step in the target image, using the CMFs of each consecutive pair of images. For each color-correction path, this approach performs *n* steps of pairwise color correction in a path with a depth of *n*. For example, let [E,C,B,F] be the path shown in Figure 5 from the target image E to reference image F, passing through images C and B. For this path, (**E**, **C**), (**C**, **B**) and (**B**, **F**) are the pairs of images that were sequentially formed along it. In this context, let ${\hat{f}}_{C\to E}$ be the CMF from source image C to target image E, let ${\hat{f}}_{B\to C}$ be the CMF from source image B to target image C and finally, let ${\hat{f}}_{F\to B}$ be the CMF from source image F to target image B. The source and target image symbolization of each CMF is only to demonstrate the color-correction order between each pair of images. Since the path of this example has a depth of 3, the color-correction procedure will perform 3 pairwise steps using the CMFs presented above. In the first step, we color correct image E using the CMF ${\hat{f}}_{C\to E}\left(E\right)$, generating the first corrected image $E^{\prime }$, which is similar in color to image C. In the second step, we color correct image $E^{\prime }$ using the CMF ${\hat{f}}_{B\to C}\left(E^{\prime }\right)$, resulting in the second-corrected image $E^{\prime \prime }$. Since the first corrected image $E^{\prime }$ already has the color similar to image C, in this step, the second corrected image $E^{\prime \prime }$ became similar in color to image B. For the third and final step for this example, we apply the CMF ${\hat{f}}_{F\to B}\left(E^{\prime \prime }\right)$ to color correct the image $E^{\prime \prime }$, producing the third and final corrected image $E^{\prime \prime \prime }$. After the three color-correction steps performed in image E, finally the final corrected image $E^{\prime \prime \prime }$ is similar in color to the reference image F. To summarize the sequential pairwise color-correction process in this example, the sequential use of the three CMFs to produce the final corrected image $E^{\prime \prime \prime }$ can be expressed as
(4)$$E^{\prime \prime \prime }={\hat{f}}_{F\to B}\left({\hat{f}}_{B\to C}\left({\hat{f}}_{C\to E}\left(E\right)\right)\right)\;.$$

Figure 8 depicts the three steps performed in this example by the sequential pairwise color correction to produce the final corrected image $E^{\prime \prime \prime }$. On the first row are the four images that composed the color correction path from target image E (in green) to reference image F (in red), from left to right. On the second row are the color-corrected images $E^{\prime }$, $E^{\prime \prime }$ and $E^{\prime \prime \prime }$, after each step of the sequential pairwise color correction.

In short, for each one of the computed paths, we perform the sequential pairwise color correction described above, producing the final corrected images, which are more similar in color to the reference image. If the target image has no color-correction path to the reference image, no color correction is performed and the image remains the same. Note that if the graph is fully connected and all the computed color-correction paths have a depth of 1, the problem is reduced to a simple pairwise color correction, meaning that each image is color corrected directly with the selected reference image. This special case occurs when all the images share a sufficient amount of photometrical information with each other, meaning that all pairs of images can be reliably used in the color-correction procedure.

The assumption behind the selection of the reference image is to have a model image that is selected using any arbitrary criterion to transform the colors of all images to be more similar to the reference one. With this arbitrary selection, the user can have control over the overall appearance of the color corrected mesh, and for that reason, we understand this approach as an advantage rather than a shortcoming of the proposed approach.

### 3.8. Image Selection for Texture Mapping

The main goal of this paper is not only to produce color-corrected images, but also to improve the visual quality of a textured mesh. To produce the textured meshes using the color-corrected images, we implemented two image selection methods: random selection and largest projection area selection. Both methods select, based on a specific criterion, one image among all available to colorize each face. This process is based on cropping, from the selected image, the projection of the face. Regarding the criterion used for each method, the random selection picks a random image among all available to colorize that face. As for the largest projection area selection, its criterion consists of selecting the image where the area of the face projection is the largest from the available ones. Furthermore, the usage of the information associated with the filtering of pairwise mappings (Section 3.2) is also important to produce high-quality textured meshes, because it assists the image-selection algorithm to avoid coloring a face with an image in which that face is occluded or with an excessively oblique viewpoint. In this context, this information is incorporated in the image-selection algorithms.

Figure 9 shows the textured meshes produced by both image-selection methods, using the original images and the color-corrected images. Figure 9a,d illustrate the image-selection methods by coloring each face with a representative color for each selected image, meaning that faces with the same color have the same selected image. The random selection method, shown in Figure 9a, uses several images to colorize the faces that represent the same surface, resulting in several texture seams artifacts. This method clearly is not the smartest approach in order to produce high-quality textured meshes; however, it facilitates the analysis of the impact of the color-corrected images in the textured mesh because it amplifies the perception of the color dissimilarities between the images. Figure 9b,c present the textured mesh produced by the random selection algorithm, using the original images and the color-corrected images, respectively. Comparing both images, the proposed approach increased considerably the visual quality of the textured mesh, eliminating most of the texture seams. As for the largest projection area selection, shown in Figure 9d, this method uses a clever criterion, producing higher-quality textured meshes with fewer image transitions throughout a surface. Figure 9e,f show the textured mesh produced by the largest projection area algorithm in the same viewpoint as for the previous algorithm, using the original images and the color-corrected images, respectively. The color-correction images combined with a better image selection algorithm clearly produced a high-quality textured mesh. Note that since the 3D mesh is a closed room, we used the back face culling method [64] to hide the faces of the mesh which are facing away from the point of view. It is possible to observe that our proposed color-correction approach produces higher-quality textured meshes, reducing significantly the number of unappealing texture seam artifacts.

## 4. Results

In this section, we analyze the influence of the proposed approach on the visual quality of the textured 3D meshes. We also compare the proposed approach with state-of-the-art approaches in quantitative and qualitative evaluations. Section 4.1 presents an image-based qualitative evaluation. Section 4.2 presents an image-based quantitative evaluation. Finally, Section 4.3 presents a mesh-based qualitative evaluation.

There are no metrics that evaluate the overall visual quality of textured meshes. As such, we base our quantitative evaluation on the assessment of how similar in color the images that are used to produce the textured meshes are. To do this, we use two image-similarity metrics to evaluate the color similarity: peak signal-to-noise ratio (*PSNR*) and *CIEDE*2000. We compute the color similarity, not between the images, but instead between pairs of associated pixels of both images. This is done using the pairwise mappings discussed in Section 3.1. To carry out a robust evaluation, the metrics use the filtered pairwise mappings, to avoid incorrect associations (see Section 3.2). The *PSNR* metric [28] measures color similarity, meaning that the higher the score values, the more similar the images. The *PSNR* metric between image **A** and image **B** can be formulated as
(5)$$\begin{array}{c} PSNR\left(A,B\right)=20*\log_{10}\left(L/RMS\right) \end{array}\;,$$
where *L* is the largest possible value in the dynamic range of an image, and RMS is the root mean square difference between the two images. The *CIEDE*2000 metric [65] was adopted as the most recent color difference metric from the International Commission on Illumination (CIE). This metric measures color dissimilarity, meaning that the lower the score values, the more similar the images. The *CIEDE*2000 metric between image **A** and image **B** can be formulated as
(6)$$\begin{array}{c} CIEDE2000\left(A,B\right)=\sqrt{{\left(\frac{\Delta L^{\prime }}{k_{L}S_{L}}\right)}^{2}+{\left(\frac{\Delta C^{\prime }}{k_{C}S_{C}}\right)}^{2}+{\left(\frac{\Delta H^{\prime }}{k_{H}S_{H}}\right)}^{2}+R_{T}\left(\frac{\Delta C^{\prime }}{k_{C}S_{C}}\right)\left(\frac{\Delta H^{\prime }}{k_{H}S_{H}}\right)} \end{array}\;,$$
where $\Delta L^{\prime }$, $\Delta C^{\prime }$ and $\Delta H^{\prime }$ are the lightness, chroma and hue differences, respectively. $S_{L}$, $S_{C}$ and $S_{H}$ are compensations for the lightness, chroma and hue channels, respectively. $k_{L}$, $k_{C}$ and $k_{H}$ are constants for the lightness, chroma and hue channels, respectively. Finally, $R_{T}$ is a hue rotation term. Since that both *PSNR* and *CIEDE*2000 are pairwise image metrics, we carry out the evaluation over all image pairs in the dataset. The displayed score values are the simple mean and the standard deviation for all image pairs. Additionally, we present a weighted mean based on the number of pairwise mappings of each pair of images. In this weighted mean, image pairs with higher number of mappings are more important to the final score value.

The qualitative evaluation of each color-correction algorithm is performed on both the images and the textured meshes. For the image-based assessment, we compare the images with respect to each other. For the mesh-based evaluation, the visual quality of the textured meshes produced by the approaches is analyzed, from different viewpoints, using both image-selection criteria discussed in Section 3.8.

Table 1 lists all the evaluated algorithms in this paper. Algorithm #1 is the baseline approach: the original images without any color correction. Algorithms #2 through #4 are pairwise color correction approaches, designed to color correct single pairs of images. As discussed in Section 2, pairwise approaches are not able to tackle the problem of the color correction of image sets. However, we presented a graph-based approach that is able to convert a multi-image color correction problem into a set of simple pairwise color correction problems. In this way, we can use classic pairwise color-correction approaches for comparison purposes. Algorithm #2 uses simple Gaussians to model the global color distribution from a source image onto a target image, in the lαβ color space [32]. Algorithm #3 estimates the CMFs using a root-polynomial regression [41]. Algorithm #4 uses the Vandermonde matrix to compute the coefficients of a polynomial that is applied as a CMF [49]. Algorithms #5 and #6 are color-correction approaches for image sets (three or more images) and are based on global optimization methods. These are both often used as baseline methods for evaluating color correction approaches for image sets [30,31,54,58,66,67]. Algorithm #5 employs an optimization method to determine a global gain compensation to minimize the color difference in the overlapped regions of the images [39]. Algorithm #6 estimates three global transformations per image, one for each color channel, by minimizing the color difference between overlapped regions through histogram matching [51]. Finally, Algorithm #7 is the proposed approach—sequential pairwise color correction approach. As described in Section 3, the threshold using the number of pairwise mappings $t_{npm}$ is the only tunable parameter for the experiments conducted in this section. We used $t_{npm}=400$, a value obtained by experimenting several possibilities and evaluating the result.

### 4.1. Image-Based Qualitative Evaluation

A qualitative evaluation of the approaches (Table 1) is presented to analyze the color similarity between the images, once the color-correction procedure is completed. After the color-correction procedure performed by each approach, it is expected that the color similarity between all images of the set has increased. Figure 10 shows the corrected images produced by each approach. Image A was selected as the reference image. We selected three other images (B, C, and D) of the 23 images available, for visualization purposes. Algorithm #1 is the baseline approach, and thus the target images B, C, and D are the original ones without any color correction. Algorithm #2 increased the contrast of the images, but did not increase the color similarity with respect to the reference image. Algorithms #3 and #4 did not produce satisfactory results since the images have degenerate colors. We believe that this is due to occlusions in the scene, which result in incorrect image correspondences. Note that these two approaches have internal image correspondence procedures, and therefore, they are not taking advantage of the filtering procedure detailed in Section 3.2. In algorithms #5 and #6, which are multi-image global optimization approaches, changes are hardly noticeable in the color corrected images, compared with the original images. The probable cause is the different sizes of the overlapped regions and the occlusions that usually occur in indoor scenarios, which produce incorrect correspondences and do not allow the optimizer to significantly reduce the color differences between all images. Another probable cause is because these algorithms use linear models, which in most of the cases are not flexible enough to deal with high color discrepancies. The proposed approach #7 produced the images with higher color similarity with respect to the reference image. Note that some images have no overlapping region with the reference image (image D), or only a small overlapping region (images B and C).

The complexity of this dataset shows that the sequential pairwise color correction, alongside a reliable pairwise mappings computation and a robust CMF estimation, played a major role to increase the color similarity between the images and the reference image. Note that this complexity is due to several images from the same scene with no overlapping regions, and several occlusions in the scene.

### 4.2. Image-Based Quantitative Evaluation

Table 2 presents the results of the image-based quantitative evaluation, for each approach. Algorithm #1 is the baseline approach, i.e., shows the level of similarity between the original images. Pairwise approaches (algorithms #2 through #4) worsened the color similarity between the images in both metrics. Those algorithms are designed to color correct only single pairs of images and, therefore, did not produce accurate results across the image set. Multi-image optimization-based algorithms #5 and #6 achieved good results and increased the color similarity between the images. However, the linear models used in these two algorithms are not flexible, and this is a probable reason why they did not achieve better results. The proposed approach #7 outperformed all other approaches. In the *PSNR* metric, we observe an improvement in the similarity between the images of around 16% (simple mean) and 13% (weighted mean) relative to the original images. In the *CIEDE*2000 metric, the color similarity is improved by around 29% in the simple mean and by around 26% in the weighted mean. These results prove that the proposed pairwise-based method is able to color correct several images from a same scene, and to achieve better results than state-of-the-art multi-image global optimization approaches.

### 4.3. Mesh-Based Qualitative Evaluation

In this section, we evaluate and compare the visual quality of the textured meshes produced by each algorithm (Table 1). As described in Section 3.8, we use two different image selection criteria to carry out the texture mapping process: random image selection and largest projection area image selection. Figure 11 points out with the red arrows the parts of the 3D mesh where the viewpoints were taken. The first viewpoint (Figure 11a) observes a corner of the room, with a high range of colors due to the posters. The second viewpoint (Figure 11b) sees a part of the room with a variety of objects (and color palettes), such as a cabinet, a whiteboard, and a few posters. Finally, the last viewpoint (Figure 11c) shows a wall that has a high luminosity from the lights of the room. The white regions of each viewpoint are holes (no triangles) in the geometry of the 3D meshes. These viewpoints show the highest difficulties of this dataset in terms of color differences. This dataset contains 24 images, and thus, the amount of transitions between images is very high, especially using the random image selection criterion. For that reason, the textured meshes using the original images contain a high number of texture seams. Figure 9a,d show where the transitions occur in the mesh, using representative colors. Figure 12 and Figure 13 show three different viewpoints using the random and largest projection area image selection criteria, respectively.

Analyzing the original images—algorithm #1 (see Figure 12, first row, and Figure 13, first row), we can see a high number of texture seams due to the color inconsistency between the images. Even with the largest projection area image selection criterion, the texture presents many visual artifacts, especially in viewpoint 3. Algorithm #2 increased the color difference between the images, resulting in even more noticeable texture seams, especially in views 1 and 3. Furthermore, as was expected, the textured meshes produced by algorithms #3 and #4 are very degenerated due to the unsuccessful color correction of the images. Algorithms #5 and #6 were also ineffective and reduced the texture seams in some specific regions, while aggravating others. For example, in Figure 12, at the top-left part of the front wall in viewpoint 3, algorithm #6 reduced the texture seams. However, on the right side, the same algorithm aggravated even more the texture seams. The proposed approach #7 outperformed all other algorithms and produced the highest-quality textured meshes with almost no texture seams, even using the random image selection criterion. These results are consistent with the image-based evaluation (see Section 4.1 and Section 4.2), that showed the proposed approach obtaining images with more similar color. These results also prove the robustness of the proposed approach, and show it is able to produce high-quality textured meshes, from datasets with many challenges, such as high amount of occlusions, high color range complexity, overlapped regions with varied sizes, and lack of overlapped regions between several pairs of images, among others.

## 5. Conclusions

The key contribution of this paper is a novel sequential pairwise color-correction approach capable of color correcting multiple images from the same scene. The sequential pairwise color-correction approach consists of selecting one image as the reference color palette model and computing a color correction path, from each image to the reference image, through a weighted graph. The color-correction weighted graph is based on the number of pairwise mappings between image pairs. Each image is color corrected by sequentially applying the CMFs of each consecutive pair of images along a color-correction path. CMFs are computed using a regression analysis called composed support vector regressor (CSVR). This procedure increases not only the color similarities of all images with respect to the reference image, but also the color similarities across all images in the set.

Results demonstrate that the proposed approach outperforms other methods in an indoor dataset using both qualitative and quantitative evaluations. More importantly, results show that the color-corrected images improved the visual quality of the textured meshes. The approach is able to correct the color differences between all images of the dataset; this is even more noticeable when using a random image selection criterion for texture mapping, a method that creates a high number of transitions between adjacent triangles. Even in this condition, the color correction results in high-quality textured meshes with little and hardly noticeable texture.

For future work, we plan to explore different metrics for the color-correction weighted graph, such as the standard deviation of a 2D Gaussian that fits the JIH data. We also intend to explore different path-selection algorithms for the color-correction paths. Furthermore, we plan to acquire larger and more complex datasets, for example, several rooms of a building, to further investigate the impact of the sequential pairwise color-correction approach, and better assess its robustness.

## Figures and Tables

**Figure 1 sensors-23-00607-f001:**
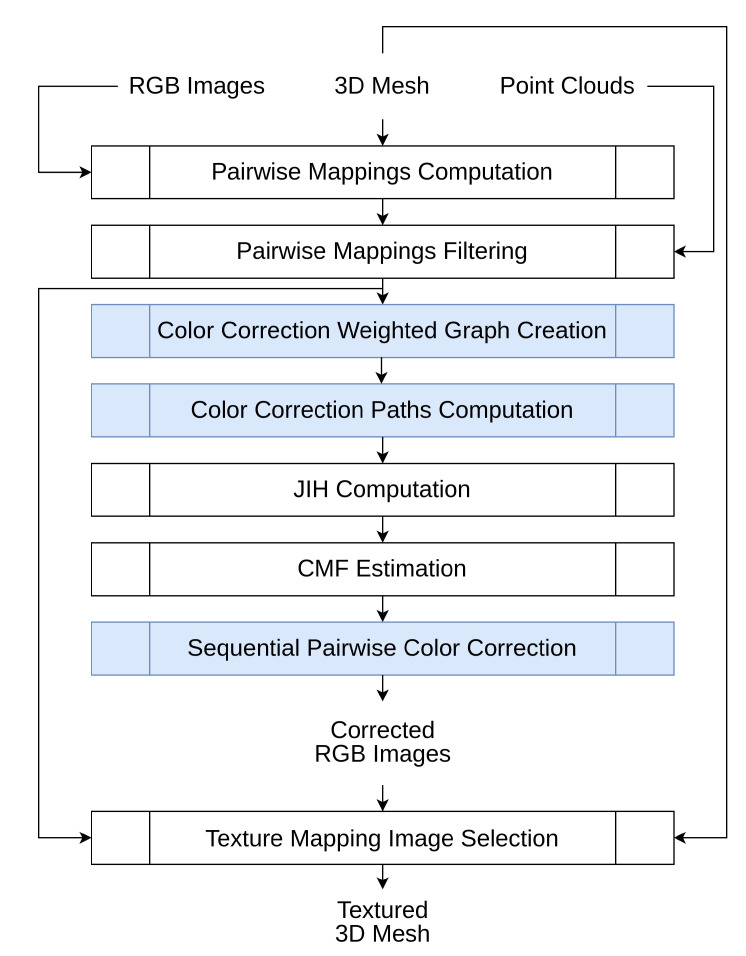
Architecture of the proposed approach. As input, our system receives RGB images, a 3D mesh, and point clouds from different scans, all geometrically registered to each other. In the first part, seven steps are performed to produce the corrected RGB images. Subsequently, we execute one last step to produce the textured 3D mesh. The core contributions of this paper are highlighted in blue. The non-squared elements represent data. The squared elements represent processes.

**Figure 2 sensors-23-00607-f002:**
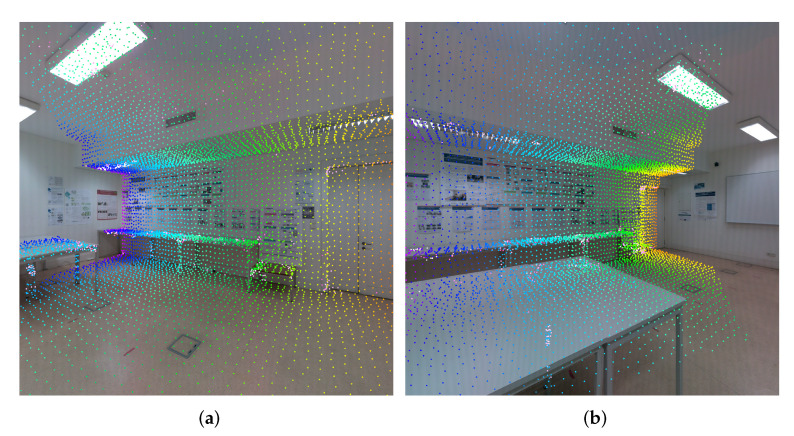
Pairwise mappings between images (**a**,**b**). The projections of the vertices are colored to identify the pairwise mappings between images. For example, in both images, the projections in orange represent the entrance door of the room.

**Figure 3 sensors-23-00607-f003:**
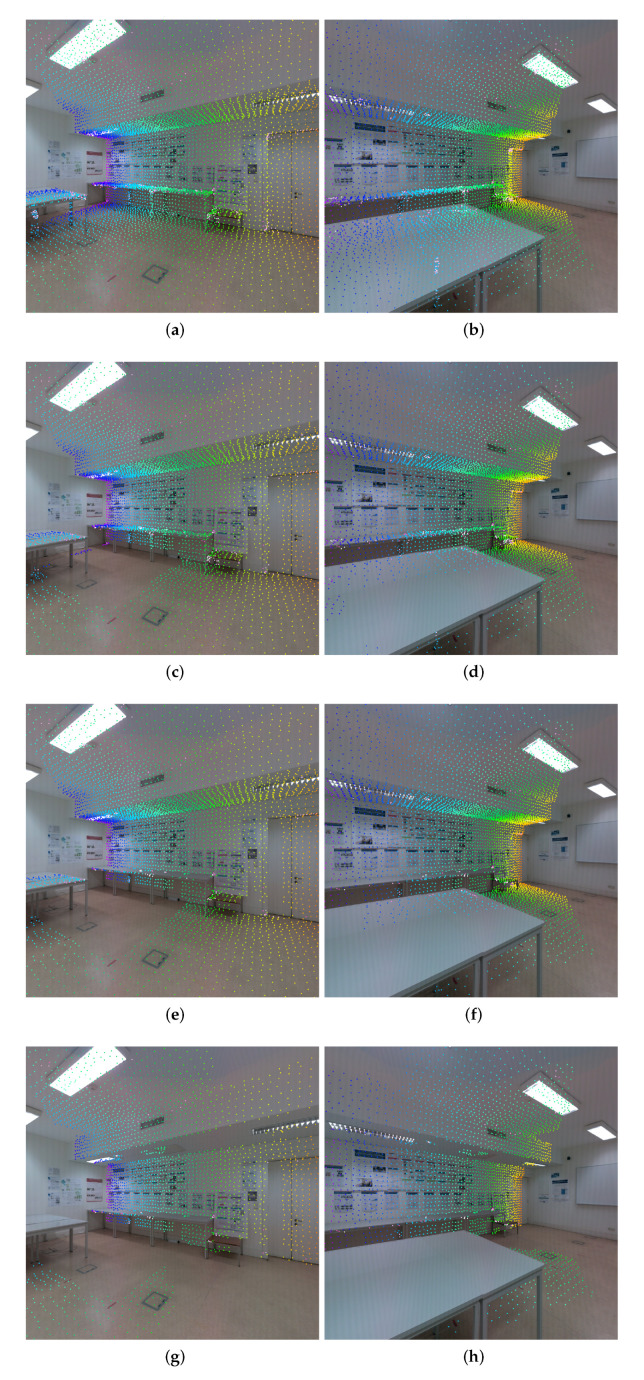
Pairwise mappings filtering procedure: (**a**,**b**) pairwise mappings with no filter applied; (**c**,**d**) pairwise mappings with z-buffering applied; (**e**,**f**) pairwise mappings with z-buffering filter and depth consistency applied; and (**g**,**h**) pairwise mappings with the entire filtering procedure applied.

**Figure 4 sensors-23-00607-f004:**
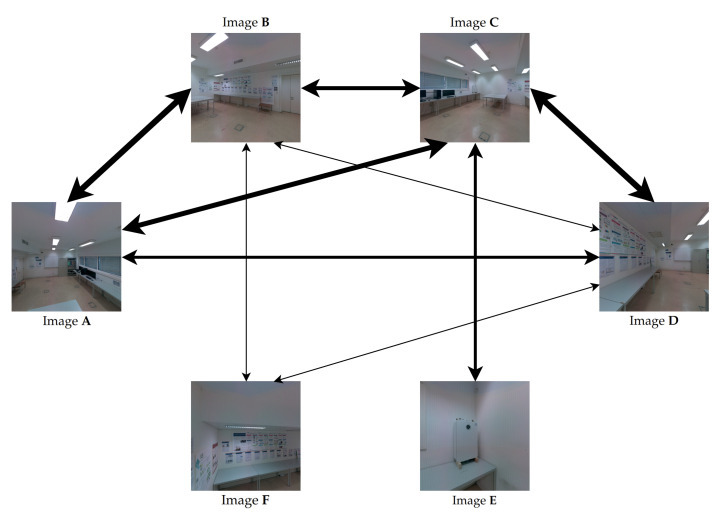
Example of a color-correction weighted graph. The nodes are represented by the individual images, the edges are represented by the arrows, and the thickness of the edges represents their weight.

**Figure 5 sensors-23-00607-f005:**
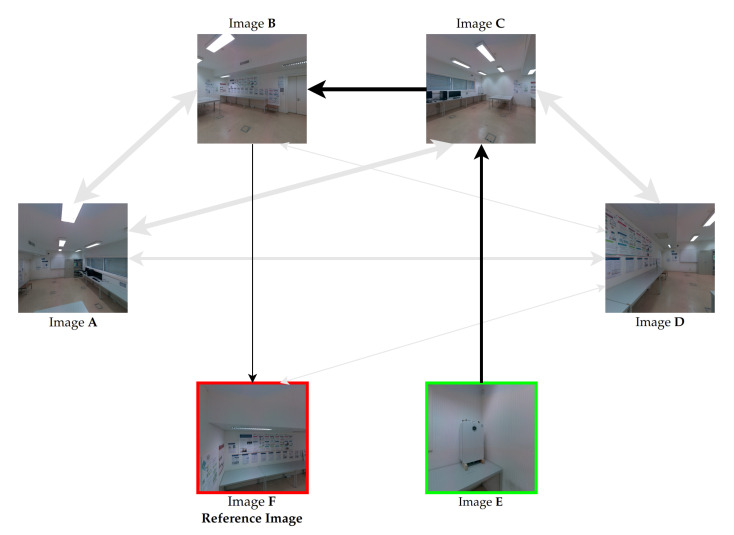
Example of the shortest path from image **E** to the reference image **F**, represented by the unidirectional black arrows. The nodes are represented by the individual images, the edges are represented by the arrows, and the thickness of the edges represents their cost. The bidirectional gray arrows are the edges of the graph that were not selected for this path.

**Figure 6 sensors-23-00607-f006:**
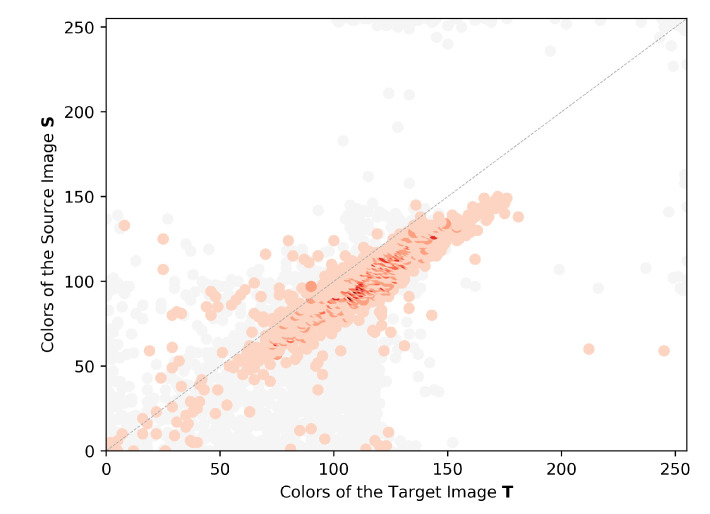
Example of a JIH of the red channel from a pair of images. The red dots represent the observations of the pairwise mappings in the red channel, and the color intensity represents the histogram count. The gray dots represent the discarded pairwise mappings.

**Figure 7 sensors-23-00607-f007:**
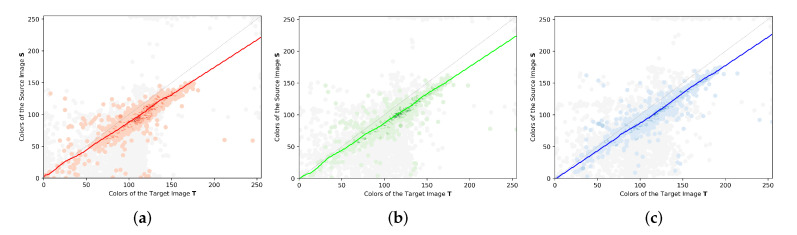
Example of the estimation of three CSVR Functions. (**a**) CMF estimation for the red channel represented by the red curve, (**b**) CMF estimation for the green channel represented by the green curve and (**c**) CMF estimation for the blue channel represented by the blue curve. The gray dots represent the discarded pairwise mappings.

**Figure 8 sensors-23-00607-f008:**
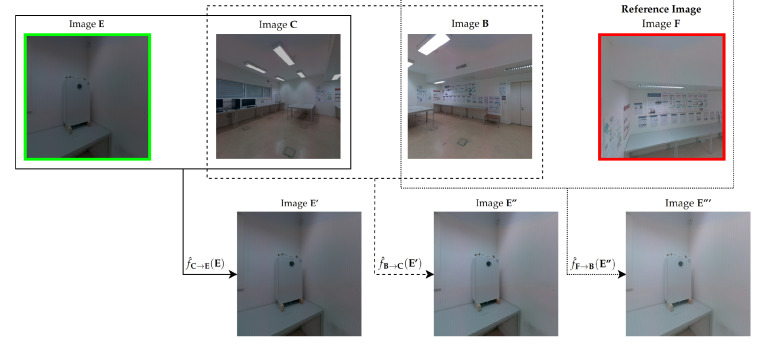
Example of the sequential pairwise color correction for a color correction path from a target image (in green) to a reference image (in red). On the first row are the four images that compose the color-correction path. On the second row are the color-corrected images, after each step of the sequential pairwise color correction. Each box (solid, dashed and dotted) represents a step of the sequential pairwise color correction, showing the CMF used and its input image. To explain our point, we drastically changed the brightness of the images on purpose, because some changes in the color of the images along the steps of the color correction procedure are not always easy to notice.

**Figure 9 sensors-23-00607-f009:**
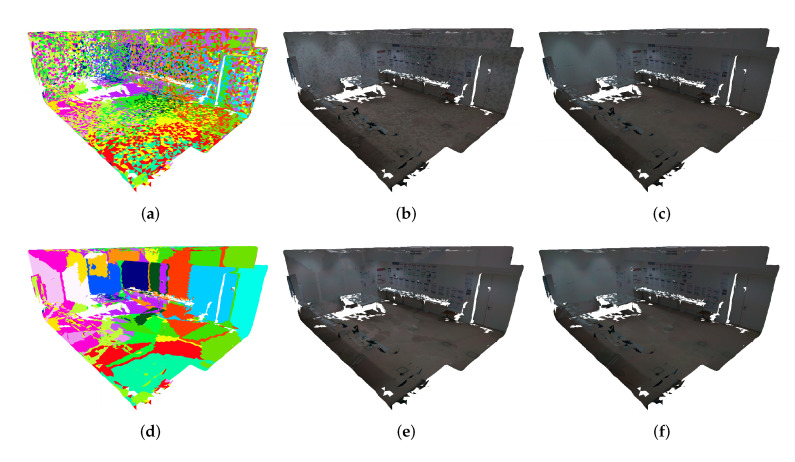
Textured meshes produced by each image selection method. (**a**,**d**) The result of the random selection and the largest projection area selection methods, respectively. (**b**,**c**) The resultant textured meshes produced by the random selection criterion using the original images and the color corrected images, respectively. (**e**,**f**) The resultant textured meshes produced by the largest projection area criterion using the original images and the color corrected images, respectively.

**Figure 10 sensors-23-00607-f010:**
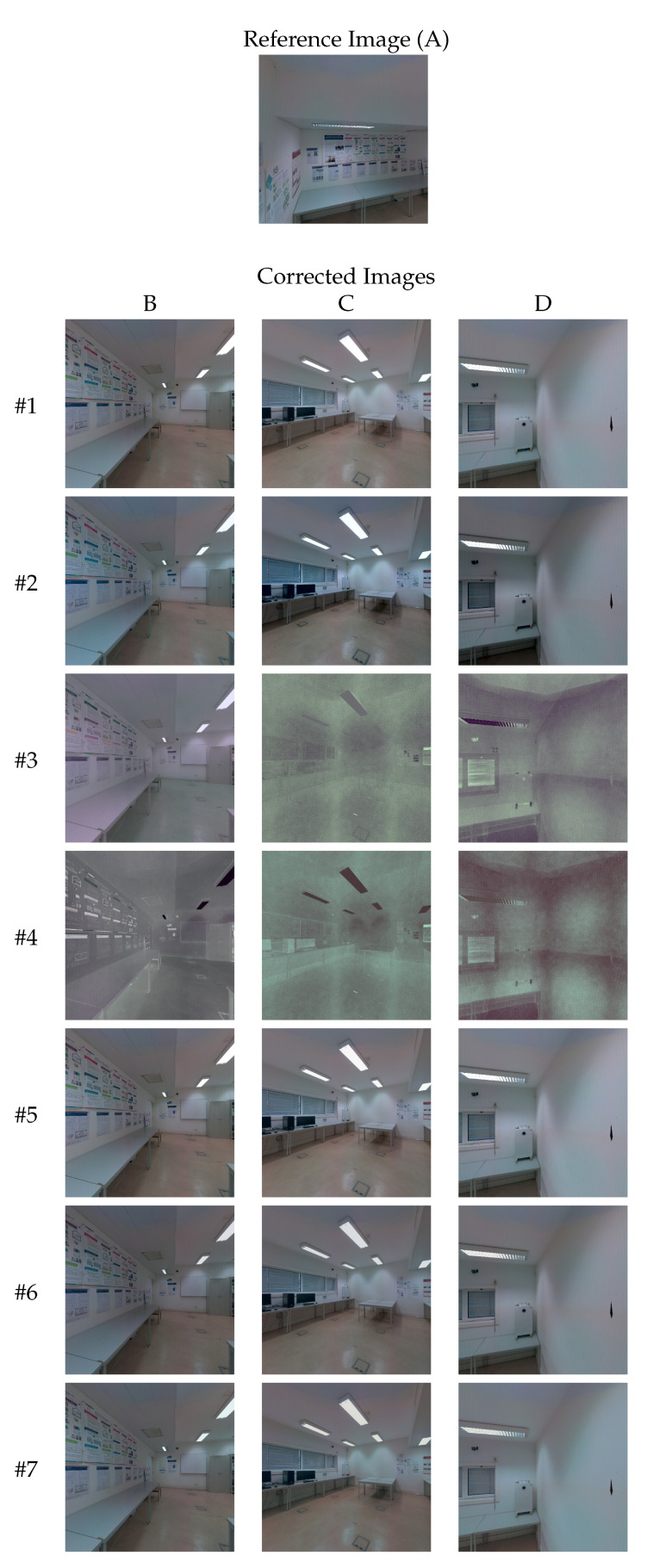
Image-based qualitative evaluation: corrected images **B**–**D** (columns) produced by each algorithm (rows), using image **A** as reference image.

**Figure 11 sensors-23-00607-f011:**
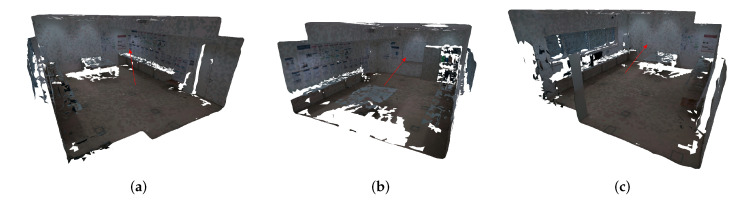
The three viewpoints analyzed in the mesh-based qualitative evaluation. The viewpoints are exemplified in the textured mesh produced by the baseline algorithm (#1) using the random image selection criterion. (**a**–**c**) Red arrows point out the parts of the 3D mesh where the viewpoints 1, 2, and 3 were taken, respectively.

**Figure 12 sensors-23-00607-f012:**
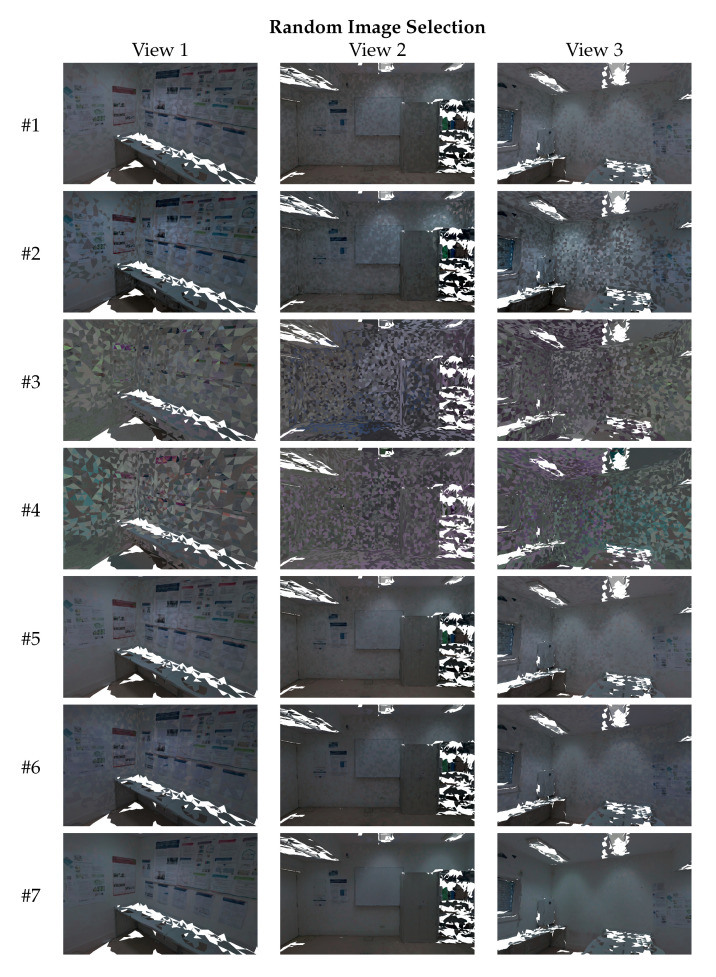
Mesh-based qualitative evaluation: textured meshes produced by each approach (rows) from three different viewpoints, and using the **random image selection technique**.

**Figure 13 sensors-23-00607-f013:**
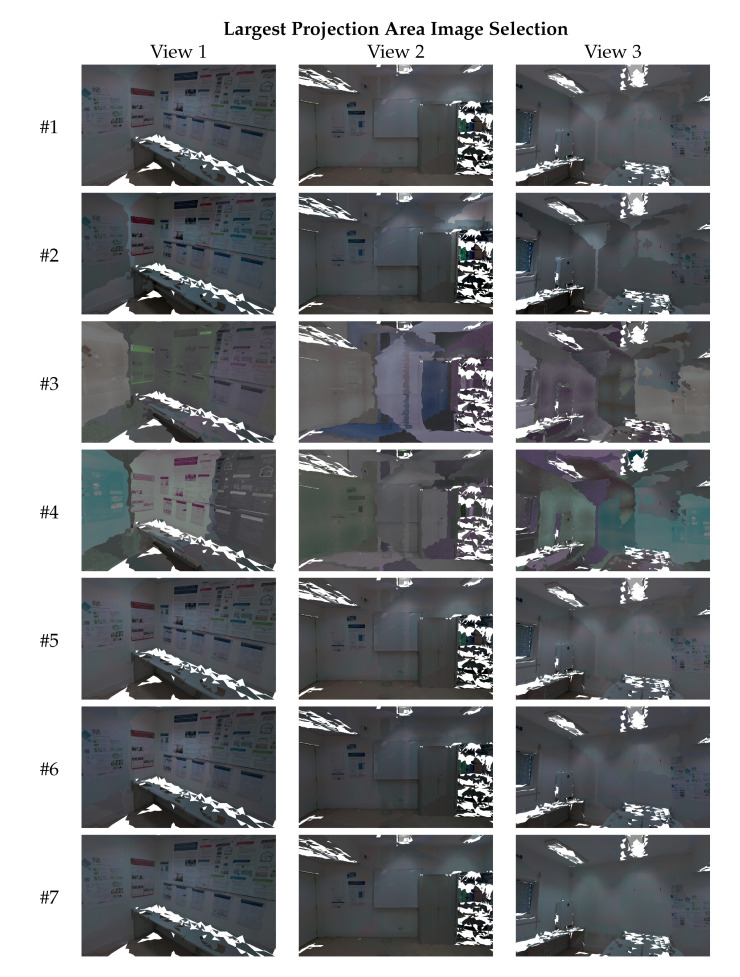
Mesh-based qualitative evaluation: textured meshes produced by each approach (rows) from three different viewpoints, and using the **largest projection area image selection technique**.

**Table 1 sensors-23-00607-t001:** State-of-the-art algorithms compared with the proposed approach.

Algorithm #	Reference	Description
#1	-	Baseline (original images)
#2	Reinhard et al. [32]	Global Color Transfer—Pairwise Approach
#3	Finlayson et al. [41]	Root-Polynomial Regression—Pairwise Approach
#4	De Marchi et al. [49]	Vandermonde Matrices—Pairwise Approach
#5	Brown and Lowe [39]	Gain Compensation—Optimization Approach
#6	Moulon et al. [51]	Histogram Matching—Optimization Approach
#7	This paper	Sequential Pairwise Color Correction

**Table 2 sensors-23-00607-t002:** Image-based quantitative evaluation: simple mean, weighted mean, and standard deviations of the *PSNR* and *CIEDE*2000 scores of all images for each algorithm (rows). The best results are highlighted in bold.

	*PSNR*			*CIEDE*2000		
Alg.	μ	$\mu_{w}$	σ	μ	$\mu_{w}$	σ
#1	24.72	24.28	3.22	6.02	5.83	1.48
#2	21.28	21.51	4.69	9.09	8.16	4.64
#3	17.33	16.74	4.46	16.28	17.30	7.18
#4	17.14	16.94	3.71	17.04	17.22	6.28
#5	26.22	25.45	3.10	5.23	5.14	1.09
#6	26.91	26.40	3.21	5.27	5.05	1.18
#7	**28.69**	**27.45**	3.51	**4.29**	**4.33**	0.83

## Data Availability

Not applicable.

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
