# Peer review of "A Sequential Color Correction Approach for Texture Mapping of 3D Meshes"

_sensors, 2023, doi:10.3390/s23020607_

Round 1

Reviewer 1 Report

The authors presented a novel methodology for color correction of multiple images from the same scene. The paper is well written, however there are some issues that should be addressed before publication.

1. The main contributions of this paper are introduced in Sections 3.3, 3.4, 3.7, as noted by the authors. These contributions/novelties should be clearly outlined.

2. The authors mix the theoretical part of the methodology with the practical part. For example, in the description of the methodology, a reader encounters the description of the input data and the results of filtering (line 360).

3. Equation 3: there is a typo at the last bracket.

4. The results were compared to 6 other algorithms, which are at least 9 years old. Why it was not compared to any newer algorithms?

Author Response

The authors presented a novel methodology for color correction of multiple images from the same scene. The paper is well written, however there are some issues that should be addressed before publication.

We would like to thank the reviewer for the positive feedback and insightful comments. Next, we reply to each of the points raised by the reviewer.

  1. The main contributions of this paper are introduced in Sections 3.3, 3.4, 3.7, as noted by the authors. These contributions/novelties should be clearly outlined.

We thank the reviewer for their suggestion, and we agree that the core contributions of this manuscript should be better outlined. Therefore, we have updated Figure 1 - Architecture of the Proposed Approach and highlighted in blue the core contributions of this paper. We also updated the caption of the figure as follows: 

“Architecture of the proposed approach. As input, our system receives RGB images, a 3D mesh, and point clouds from different scans, all geometrically registered to each other. In the first part, seven steps are performed to produce the corrected RGB images. Subsequently, we execute one last step to produce the textured 3D mesh. The core contributions of this paper are highlighted in blue. The non-squared elements represent data. The squared elements represent processes.”

Additionally, we have updated the abstract to clearly outline the novelties of this paper as follows:

“Texture mapping can be defined as the colorization of a 3D mesh using one or multiple images. In the case of multiple images, this process often results in textured meshes with unappealing visual artifacts, known as texture seams, caused by the lack of color similarity between the images. The main goal of this work is to create textured meshes free of texture seams by color correcting all the images used. We propose a novel color correction approach, called sequential pairwise color correction, capable of color correcting multiple images from the same scene, using a pairwise-based method. This approach consists of sequentially color correcting each image of the set with respect to a reference image, following color correction paths computed from a weighted graph. The color correction algorithm is integrated with a texture mapping pipeline that receives uncorrected images, a 3D mesh, and point clouds as inputs, producing color corrected images and a textured mesh as outputs. Results show that the proposed approach outperforms several state-of-the-art color correction algorithms, both in qualitative and quantitative evaluations. The approach eliminates most texture seams, significantly increasing the visual quality of the textured meshes.”

  1. The authors mix the theoretical part of the methodology with the practical part. For example, in the description of the methodology, a reader encounters the description of the input data and the results of filtering (line 360).

We believe the reviewer is mentioning the information about the percentages of pairwise mappings removed by each filter, which are measures specific to this dataset and will be different for other datasets. Our intention was to show the usefulness of the proposed filter with statistics from a real case study, but we agree with the reviewer in that it is confusing to do it at this stage. We removed the following sentences about these percentages: 

“This filtering was capable to eliminate 30.03% of inaccurate pairwise mappings due to occluded faces.”

“This filtering solely was capable to eliminate an additional 9.96% of inaccurate pairwise mappings due to depth inconsistencies.”

“Finally, this filtering solely was capable to eliminate an additional 22.12% of inaccurate pairwise mappings.”

“The entire filtering procedure led to the removal of 62.11% of the pairwise mappings, which clearly demonstrates the extensive amount of noise in the dataset used.”

Additionally, we also rephrased the last sentence to conclude Section 3.2 with the following: 

“Figure 3 clearly shows the extensive amount of noise in the dataset used. This reinforces the importance of using 3D information not only to compute the correspondences between images, but to discard the incorrect ones.”

In Section 3.6 - Estimation of Color Mapping Functions, we also mention the percentage of pairwise mappings the filters removed from a pair of images in specific. Accordingly, we also rephrased the following sentence:

“The amount of noisy mappings (gray dots) eliminated by the filtering procedure is very high, and only for this pair of images, the filters were able to discard 67.94\% of the pairwise mappings.”

To:

“The amount of noisy mappings (gray dots) eliminated by the filtering procedure is significant (see Figure 6 and Figure 7).” 

  1. Equation 3: there is a typo at the last bracket.

We agree and thank the reviewer for pointing this out. Accordingly, we have changed equation 3.

  1. The results were compared to 6 other algorithms, which are at least 9 years old. Why it was not compared to any newer algorithms?

As discussed in Section 2, there are two major alternatives to address the problem of color correction. On the one hand, there are pairwise-based color correction methods, while on the other hand, there are optimization-based methods. Optimization-based methods are easier to use for image sets when compared to pairwise-based methods. Perhaps because of this limitation, the state of the art on pairwise-based methods has not advanced in the last 5 years. Nonetheless, pairwise-based methods do have some advantages when compared to optimization approaches, in particular the fact that they are simpler and more efficient. This paper proposes a graph-based approach that is able to generalize pairwise-based methods for color correcting image sets. Despite the recent attention to other methods, such as neural networks, the fact is that as demonstrated in [1], these novel algorithms do not outperform the regression-based algorithms, such as the root-polynomial regression used in algorithm #3 - Finlayson et al. 

In this context, pairwise-based methods remain the state of the art, which is why we decided it was important to compare our approach against other pairwise-based color correction methods.  Furthermore, since we leverage the state of the art of color correction for image pairs to color correct multiple images from the same scene, we also compared our pairwise-based approach with two color correction approaches for image sets, algorithms #5 (Brown and Lowe 2007) and #6 (Moulon et al. 2013). The reason we chose these two algorithms is that they are the most used papers for comparisons and have a significant number of citations. In other words, these are established baseline methods used for comparison in very recent proposals [2-7].  In short, we believe these 5 methodologies to which we compare are representative and often used as the baseline standard. The implementation of these algorithms is not open source, and we had to implement all of these papers in a reasonable amount of time to be able to compare them with our algorithm.

Considering the reviewer’s comment, we decided to add this explanation in the paper and added the following sentence to Section 4 - Results: 

“These are both often used as baseline methods for evaluating color correction approaches for image sets [30,31,54,58,66,67].”

[1] Kucuk, A., Finlayson, G., Mantiuk, R., & Ashraf, M. (2022). Comparison of Regression Methods and Neural Networks for Colour Corrections. London Imaging Meeting, 3(1), 74–79. https://doi.org/10.2352/lim.2022.1.1.1

[2] Xiong, Y., & Pulli, K. (2010). Color and luminance compensation for mobile panorama construction. Proceedings of the International Conference on Multimedia - MM ’10, June, 1547. https://doi.org/10.1145/1873951.1874281

[3] Xia, M., Yao, J., & Gao, Z. (2019). A closed-form solution for multi-view color correction with gradient preservation. ISPRS Journal of Photogrammetry and Remote Sensing, 157(May), 188–200. https://doi.org/10.1016/j.isprsjprs.2019.09.004

[4] Li, L., Xia, M., Liu, C., Li, L., Wang, H., & Yao, J. (2020). Jointly optimizing global and local color consistency for multiple image mosaicking. ISPRS Journal of Photogrammetry and Remote Sensing, 170(October), 45–56. https://doi.org/10.1016/j.isprsjprs.2020.10.006

[5] Li, Y., Yin, H., Yao, J., Wang, H., & Li, L. (2022). A unified probabilistic framework of robust and efficient color consistency correction for multiple images. ISPRS Journal of Photogrammetry and Remote Sensing, 190(June), 1–24. https://doi.org/10.1016/j.isprsjprs.2022.05.009

[6] Shen, T., Wang, J., Fang, T., Zhu, S., & Quan, L. (2017). Color correction for image-based modeling in the large. In Lecture Notes in Computer Science (including subseries Lecture Notes in Artificial Intelligence and Lecture Notes in Bioinformatics): Vol. 10114 LNCS (pp. 392–407). https://doi.org/10.1007/978-3-319-54190-7_24

[7] Yang, J., Liu, L., Xu, J., Wang, Y., & Deng, F. (2021). Efficient global color correction for large-scale multiple-view images in three-dimensional reconstruction. ISPRS Journal of Photogrammetry and Remote Sensing, 173(December 2020), 209–220. https://doi.org/10.1016/j.isprsjprs.2020.12.011

Reviewer 2 Report

1、 The experimental analysis diagram is not clear enough. The author can replace it with a clearer picture.
2、 Proper nouns should be explained more clearly to help readers understand such as JIH
3、 In the qualitative quality assessment based on images, we see that there is no obvious visual difference between the method proposed in this paper and the method 5 and 6. Please explain the progressiveness of this method.
4、 The meaning of color correction in the introduction is not clear enough, please clearly state the meaning of correction.

Author Response

We would like to thank the reviewer for the insightful comments, which have been very helpful for improving the manuscript. Next, we reply to each of the points raised by the reviewer.

  1. The experimental analysis diagram is not clear enough. The author can replace it with a clearer picture.

We believe the reviewer is referring to Figure 1 - Architecture of the Proposed Approach.  We agree with the reviewer that this figure is not clear enough. Accordingly, we have replaced this figure and have changed the layout of the architecture to make it more clear and more understandable.

With this new layout, the core contributions of this paper are highlighted in blue. The non-squared elements represent data. The squared elements represent processes.

Additionally, we believe it is a good practice to explicitly describe the meaning of each element in the caption of the figure. In this light, we updated the caption of Figure 1 as follows:

“Architecture of the proposed approach. As input, our system receives RGB images, a 3D mesh, and point clouds from different scans, all geometrically registered to each other. In the first part, seven steps are performed to produce the corrected RGB images. Subsequently, we execute one last step to produce the textured 3D mesh. The core contributions of this paper are highlighted in blue. The non-squared elements represent data. The squared elements represent processes.”

  1. Proper nouns should be explained more clearly to help readers understand such as JIH

We thank the reviewer for pointing out some terms that are not clearly explained. In this context, we have conducted an extensive revision of all terms explained in the manuscript to make sure that these are clearly explained to help the readers to understand them. For example, we added the following sentence in Section 2 - Related Work to introduce the term Joint Image Histogram (JIH): 

“A JIH is a 2D histogram that shows the relationship of color intensities at the exact position between a target image T, and a source image S.”

  1. In the qualitative quality assessment based on images, we see that there is no obvious visual difference between the method proposed in this paper and the method 5 and 6. Please explain the progressiveness of this method.

We agree with the reviewer that, in this case, there is no noticeable visual difference in the images between our method (#7) and methods #5 and #6, regarding Section 4.1 - Image-based Qualitative Evaluation. In fact, visual comparisons are important but have limitations. That is why we also present quantitative metrics to measure the color similarity between the images. In this light, Section 4.2 - Image-based Quantitative Evaluation demonstrates that our proposed approach outperformed all other algorithms, including algorithms #5 and #6. Table 2 shows quantitatively that the images are more similar in color to each other after the color correction performed by our approach, according to the image similarity metrics PSNR and CIEDE2000. Furthermore, 4.3 - Mesh-based Qualitative Evaluation compares the textured meshes from different points of view. Figures 12 and 13 show that our approach improved the visual quality of the textured meshes and produced the highest-quality textured meshes among the compared approaches in both image selection criteria. These results reinforce that our approach outperformed all other algorithms, including algorithms #5 and #6. 

In short, a visual comparison is helpful but must always be confirmed by a quantitative evaluation because it is hard to visually perceive color discrepancies in two separate images. However, if these images are used for texture mapping a mesh, minor color discrepancies which are difficult to identify through a visual inspection of the image, may create visual seams in the mesh texture.

  1. The meaning of color correction in the introduction is not clear enough, please clearly state the meaning of correction.

We agree with the reviewer that the meaning of color correction is not clear enough in the introduction, and we thank the reviewer for paying such close attention. Accordingly, we added the following sentence in Section 1 - Introduction to clearly state the meaning of color correction: 

“Color correction can be defined as the general problem of compensating the photometrical disparities between two coarsely geometrically registered images. In other words, color correction consists of transferring the color palette of a source image S to a target image T [28].“

Reviewer 3 Report

This work focused on a sequential color correction approach for texture mapping of 3D meshes. It is an teresting and novel application about sensors. The paper is well organized and the results are well demonstrated. I can express some limited support for this work, and hope it can make some minor improvement.

1. There are too many short paragraphs in manuscript. It is not usual and uniformed. Please deal with it.

2. The novelty of this paper needs to be better illustrated and described, especially in Abstract.

3. The basic parameter setting for experiments need to be listed in a Table.

4. The evaluation metrics need to be described in this paper, as there are some readers who cannot well know the two metrics.

5. The computer vision area is a rapidly-developing area. Please read some new papers to absorb more research impression. Such as:

- Integration of Light Curve Brightness Information and Layered Discriminative Constrained Energy Minimization for Automatic Binary Asteroid Detection

- DeformableGAN: Generating Medical Images With Improved Integrity for Healthcare Cyber Physical Systems

Author Response

This work focused on a sequential color correction approach for texture mapping of 3D meshes. It is an interesting and novel application about sensors. The paper is well organized and the results are well demonstrated. I can express some limited support for this work, and hope it can make some minor improvement.

We would like to thank the reviewer for the positive feedback and insightful comments, which have been very helpful for improving the manuscript. Next, we reply to each of the points raised by the reviewer.

  1. There are too many short paragraphs in manuscript. It is not usual and uniformed. Please deal with it.

We agree with the reviewer that there are too many short paragraphs in the manuscript. Thank you for pointing it out. Accordingly, we have conducted an extensive revision of the manuscript to solve this issue, and we believe that now the manuscript is more uniform and well-organized. 

  1. The novelty of this paper needs to be better illustrated and described, especially in Abstract.

We thank the reviewer for their suggestion. Accordingly, we have updated the abstract to better describe the novelty of this paper. The updated abstract is presented below:

“Texture mapping is defined as the colorization of a 3D mesh using a single or multiple images. In the case of multiple images, this process often results in textured meshes with unappealing visual artifacts known as texture seams, caused by the lack of color similarity between the images. The main goal of this work is to create textured meshes free of texture seams by color correcting all images available from the scene. We propose a novel color correction approach, called sequential pairwise color correction, capable of color correcting multiple images from the same scene, using a pairwise-based method. This approach consists of sequentially color correcting each image of the set w.r.t. a reference image, using color correction paths computed from a color correction weighted graph. The color correction algorithm is integrated with a texture mapping pipeline that receives uncorrected images, a 3D mesh, and point clouds as inputs, and produces color corrected images and a textured mesh as outputs. Results show that the proposed approach outperforms several state-of-the-art color correction algorithms, both in qualitative and quantitative evaluations. The approach eliminates most texture seams, significantly increasing the visual quality of the textured meshes.”

The new contributions of this paper were also highlighted in blue in Figure 1 - Architecture of the Proposed Approach.

  1. The basic parameter setting for experiments need to be listed in a Table.

As described in Section 3 - Proposed Approach, the only tunable parameter is the threshold using the number of pairwise mappings to create edges between pairs of images in the weighted graph. As such, we do not think it is helpful to create a table that only evaluates the impact of this single parameter. To the reviewer’s point, we have added a sentence at the beginning of Section 4 - Results to present this basic parameter setting:

“As described in Section 3, the threshold using the number of pairwise mappings Tnpmis the only tunable parameter for the experiments conducted in this Section. We used  Tnpm = 400, a value obtained by experimenting several possibilities and evaluating the result. 

  1. The evaluation metrics need to be described in this paper, as there are some readers who cannot well know the two metrics.

We agree with the reviewer that the evaluation metrics are not well described in the manuscript. As such, we added Equations 5 and 6 to formally introduce these image similarity metrics as follows: 

“The PSNR metric [28] measures color similarity, meaning that the higher the score values, the more similar are the images. The PSNR metric between image A and image B can be formulated as:

see manuscript for equation

where L is the largest possible value in the dynamic range of an image, and RMS is the root mean square difference between the two images. The CIEDE2000 metric [65] was adopted as the most recent color difference metric from the International Commission on Illumination (CIE). This metric measures color dissimilarity, meaning that the lower the score values, the more similar are the images. The CIEDE2000 metric between image A and image B can be formulated as:

see manuscript for equation

where ∆L′, ∆C′ and ∆H′ are the lightness, chroma and hue differences, respectively. SL, SC and SH are compensations for the lightness, chroma and hue channels, respectively. kL, kC and kH are constants for the lightness, chroma and hue channels, respectively. Finally, RT is a hue rotation term.“

  1. The computer vision area is a rapidly-developing area. Please read some new papers to absorb more research impression. Such as:

- Integration of Light Curve Brightness Information and Layered Discriminative Constrained Energy Minimization for Automatic Binary Asteroid Detection

- DeformableGAN: Generating Medical Images With Improved Integrity for Healthcare Cyber Physical Systems

We thank the reviewer for indicating these references. We read through them and found that one of them is related to our work, and we included it in the manuscript.

Round 2

Reviewer 1 Report

The authors have provided adequate responses and revised the paper accordingly.

Reviewer 3 Report

The reviewer is satisfied with the revision.